# Reasoning to Regulate: Chain-of-Thought for Traffic Rule Understanding

## Abstract

Understanding and complying with traffic regulations is a safety-critical requirement for autonomous driving, yet remains challenging due to the diversity and context dependence of traffic signage. Importantly, regulation understanding is not a simple recognition task, but a reasoning problem: whether a rule applies depends on interpreting the sign in relation to the spatial layout of lanes and scene context. To support such reasoning, MapDR provide fine-grained annotations that link each traffic sign's regulatory rules to the specific lanes they govern. Existing methods, however, largely treat this as direct sequence prediction, ignoring the underlying reasoning that connects sign semantics and map structure. To address this limitation, we explicitly incorporate reasoning into this task and propose a framework that equips vision-language models (VLMs) with chain-of-thought (CoT) capabilities. We first design a scalable CoT curation pipeline that bootstraps rationales from a strong LLM through a two-round strategy and employs a VLM-based verifier to filter out incorrect cases, yielding a high-quality set of (CoT, answer) pairs. Building on this foundation, we adopt a two-stage training scheme: supervised fine-tuning (SFT) to teach rationale-to-answer generation, followed by GRPO reinforcement learning with answer-grounded, fine-grained rewards to further improve final answer accuracy. Extensive experiments on MapDR show that our approach significantly improves both interpretability and accuracy, establishing the first reasoning-based framework for regulation-aware autonomous driving.

## 1 Introduction

Autonomous agents, particularly autonomous driving systems, must comply with traffic regulations while driving. Violations of traffic rules can lead to severe safety risks; thus, accurately understanding traffic regulations and assessing their applicability to the ego vehicle is crucial. In practice, mapping traffic rules to specific lanes requires not only semantic comprehension of signs but also complex contextual reasoning, making the task highly challenging.

Recent datasets have started to couple perception with the prediction of relationships between lanes and traffic elements such as traffic lights and signs. **OpenLane-V2** Wang et al. (2023a) as shown in Figure 1(a) augments classic HD map benchmarks such as *nuScenes* Caesar et al. (2020) and *Argoverse* Wilson et al. (2023) with Lane-to-Lane and Lane-to-Traffic-element associations, where the latter links traffic elements (e.g., signs, signals) to specific lanes. However, OpenLane-V2 mainly focuses on directional signs and provides only category-level annotations, which limits its applicability in real-world scenarios where traffic signs are diverse and context-dependent, and lacks the fine-grained rule descriptions required for safe decision making. **MapDR** Chang et al. (2025) advances this line of work by providing detailed annotations of traffic sign attributes and their connections to lane-level applicability, thereby enabling the study of regulation-aware driving on top of online HD maps, as illustrated in Figure 1(b).

However, developing effective algorithms on the MapDR dataset remains largely underexplored and challenging. A key difficulty lies in emulating human-like reasoning to enable agents to understand traffic regulations. Real-world traffic signage is highly diverse and compositional, and its applicability to a given lane depends on contextual factors such as spatial topology and surrounding context. Prior work Chang et al. (2025) addresses this by formulating the task as direct sequence generation, predicting the final answer token by token from observations (front-view image, cropped traffic sign,

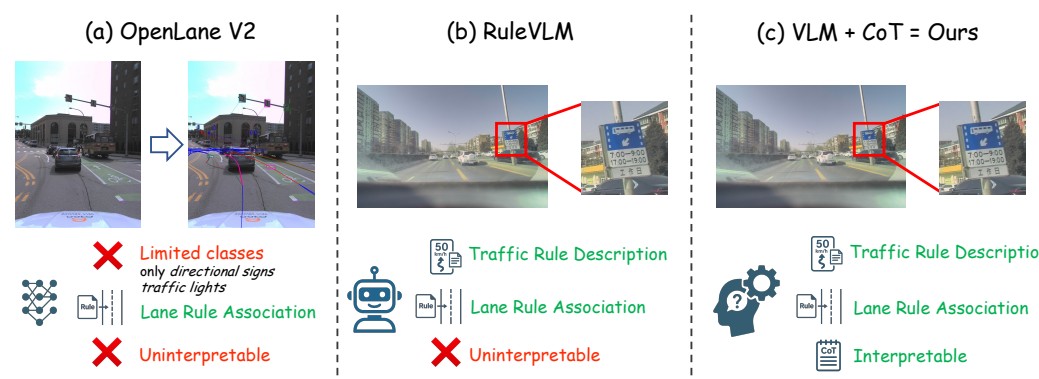

Figure 1: Comparative illustration of three paradigms in traffic rule understanding and lane-rule association. (a) The OpenLane V2 style vision-driven paradigm handles limited sign classes (e.g. directional signs, traffic lights) and lacks reasoning process, thus being largely uninterpretable. (b) RuleVLM performs end-to-end traffic rule description and attempts to map rules to lanes, but remains a black-box ("uninterpretable") model without revealing its reasoning process. (c) Our VLM+CoT approach integrates Chain-of-Thought reasoning: it can generate traffic rule descriptions, map them to specific lanes, and expose its intermediate reasoning, making the model more interpretable and its decisions more explainable.

and lane information). However, this formulation bypasses the reasoning process underlying rule interpretation and under-utilizes the latent reasoning capacity of VLMs.

To further improve performance, we argue that explicit reasoning is necessary to emulate the human inference process. Rather than directly predicting the final answer, we train a VLM to perform reasoning and base its decision on thinking process. Concretely, the model (i) generates a chain-of-thought (CoT) Wei et al. (2022) that explains how a traffic rule applies (or does not apply) to the target lane, and (ii) derives the final decision from this rationale.

Concretely, we first design a **CoT curation pipeline** that bootstraps rationales from a strong LLM and validates them against ground-truth labels. In the first stage, the model is prompted to generate both rationales and answers; we then compare its predictions with the annotations and request the model to revise its rationale based on correctness. In the second stage, a VLM-based verifier uses the generated CoT to infer the final answer and checks its consistency with ground truth, thereby filtering out incorrect or ungrounded rationales. This process yields a high-precision set of CoT-annotated samples, which are subsequently used to train models with both strong reasoning capability and reliable decision accuracy. During training, to equip the model with CoT reasoning ability, we adopt a **two-stage strategy**. We first perform supervised fine-tuning (SFT) to teach the VLM to generate *(rationale → answer)* sequences. While SFT allows the model to follow the protocol and learn basic reasoning traces, it often yields suboptimal final answers. To overcome this limitation, we further apply GRPO Shao et al. (2024) with an answer-grounded, fine-grained reward, where the model samples multiple reasoning paths and is reinforced towards those that produce correct outputs.

We summarize our contributions as follows:

- To the best of our knowledge, we are the first to introduce CoT reasoning into traffic regulation understanding, moving beyond direct sequence prediction, which is critical for safety-sensitive autonomy.

- We propose a scalable CoT data curation pipeline that combines a two-round generation strategy with self-checking and VLM-based verification, enabling the reliable construction of large-scale (CoT, answer) pairs.

- We adopt a two-stage training schedule (SFT → GRPO) with *answer-grounded, fine-grained* rewards to enhance reasoning quality and decision accuracy.

- We conduct extensive experiments and ablation studies, demonstrating the effectiveness of our approach.

## 2 RELATED WORKS

### 2.1 LANE–RULE ASSOCIATION

3D Lane and road-topology recognition is critical for safe autonomous driving Chen et al. (2022); Luo et al. (2023; 2024b); Wang et al. (2023b); Huang et al. (2024); Ma et al. (2024). Building on this, OpenLane-V2 Wang et al. (2023a) establishes a large-scale topology reasoning benchmark that links lanes and traffic elements, spurring follow-up methods on lane–lane and lane–traffic reasoning Wu et al. (2023a); Fu et al. (2024); Li et al. (2023). However, OpenLane-V2 constrained by single-label classification for traffic sign rather than structured descriptions for fine-grained driving rules. This makes it insufficient for signs with multiple rules, which are complex but common in real scenarios. MapDR addresses this gap by focusing on driving-rule extraction from traffic signs and lane-level association Chang et al. (2025). However, OpenLane-V2 remains limited to single-label categorization of traffic elements and their lane links, rather than structured, fine-grained rule descriptions. MapDR addresses this gap by introducing a benchmark for driving-rule extraction from traffic signs and lane-level association. While RuleVLM Chang et al. (2025) treats the task as direct sequence prediction of the final answer, it overlooks the task's inherent need for explicit, compositional reasoning.

### 2.2 CHAIN-OF-THOUGHT

LLMs have demonstrated remarkable emergent capabilities in complex reasoning tasks Talmor et al. (2018); Roy & Roth (2016). Among these, chain-of-thought (CoT) prompting significantly improves reasoning performance, whether via manually crafted exemplars Wei et al. (2022) or via zero-shot self-rationalization prompts like "Let's think step by step" Kojima et al. (2022). To scale CoT, Auto-CoT Zhang et al. (2022) automatically samples diverse questions and generates reasoning chains to build large demonstration sets without human effort. Beyond prompting methods, recent work investigates how to *distill* CoT reasoning from large teacher models into smaller student models (SLMs)[1]. Chen et al. (2025b) conduct empirical study and uncover three key insights in CoT distillation: 1) simpler CoTs can outperform finer ones for SLMs; 2) CoT format exerts minimal effect on SLMs; 3) diversity and complexity in the rationale set often matter more. Meanwhile, Liu et al. (2023) argue that self-instruction tuning methods often underperform on complex reasoning tasks,introduce LOGICOT, a chain-of-thought instruction-tuning dataset to improves performance. In evaluation, the paradigm of *LLM-as-a-judge* has become a practical approach for judging faithfulness, coherence, and step validity. Recent surveys examine the reliability challenges and design strategies for LLM judges Li et al. (2025a), while other works use attribute-wise assessments or prompt judges to produce rich quality judgments Zhang et al. (2024); Guo et al. (2024). In particular, for mathematical reasoning, some frameworks go beyond final accuracy and evaluate individual reasoning steps for validity, coherence, or redundancy via dedicated judge LLMs Xia et al. (2025).

### 2.3 LLM/VLM FOR AUTONOMOUS DRIVING

Recent work leverages LLMs/VLMs for autonomous driving; a particularly important direction is end-to-end driving and planning Hwang et al. (2024); Xu et al. (2024b); Wang et al. (2025); Sima et al. (2024); Chen et al. (2024); Cao et al. (2024); Li et al. (2025b); Xu et al. (2024a). Within end-to-end driving, incorporating chain-of-thought (CoT) reasoning has emerged as an important research direction. DriveCoT Wang et al. (2024) incorporates CoT reasoning into end-to-end driving, offering a CARLA-based dataset with rationale annotations and a baseline agent that predicts both rationales and control decisions. PKRD-CoT Luo et al. (2024a) proposes a zero-shot CoT prompting framework that structures autonomous-driving reasoning into four stages: *Perception*, *Knowledge*, *Reasoning*, and *Decision*. DriveVLM Sima et al. (2024) organizes end-to-end driving as a three-stage CoT: scene description, scene analysis, and hierarchical planning, targeting long-tail scenarios. EMMA Hwang et al. (2024) builds an end-to-end, language-centric driving model that maps multi-camera images to planner trajectories, 3D objects, and road-graph elements. RoboTron-Drive Huang et al. (2025) augments and standardizes multiple autonomous driving datasets to fine-tune a unified large multimodal model, resulting in an all-in-one LMM that supports perception, prediction, and planning. Sce2DriveX Zhao et al. (2025) leverages joint learning from local scene

---

[1] https://huggingface.co/blog/jjokah/small-language-model

videos and global BEV maps to capture long-range spatiotemporal relations and road topology. CoT-Drive Liao et al. (2025) distills LLM reasoning into lightweight edge models and leverages CoT prompting to enable efficient, accurate, real-time motion forecasting. AlphaDrive Jiang et al. (2025) integrates GRPO-based reinforcement learning with planning-specific rewards and explicit reasoning training into VLM-based end-to-end driving. AgentThink Qian et al. (2025) introduces a structured data generation pipeline and a two-stage training strategy that integrates chain-of-thought reasoning with dynamic, agent-style tool invocation for autonomous driving tasks.

## 3 METHOD

In this section, we present our method in detail. The overall framework is illustrated in Figure 2, and can be divided into four components: *CoT Data Generation*, *CoT Data Filter*, *Supervised Fine-Tuning*, and *Reinforcement Fine-Tuning*. The first two components are responsible for generating and filtering high-quality CoT datasets, while the latter two progressively train the VLM with reasoning ability.

We first elaborate our proposed CoT data curation pipeline in Section 3.1, explaining how we adopt a two-round strategy to prompt a VLM to generate (CoT, answer) pairs and how we employ a second VLM to filter out incorrect rationales. Then, in Section 3.2, we describe our two-stage training pipeline: 1) a supervised fine-tuning (SFT) stage trained on mixed CoT and non-CoT (answer-only) data, and 2) a reinforcement fine-tuning (RFT) stage using GRPO with a fine-grained, answer-grounded reward to refine the model's reasoning and rule–lane alignment.

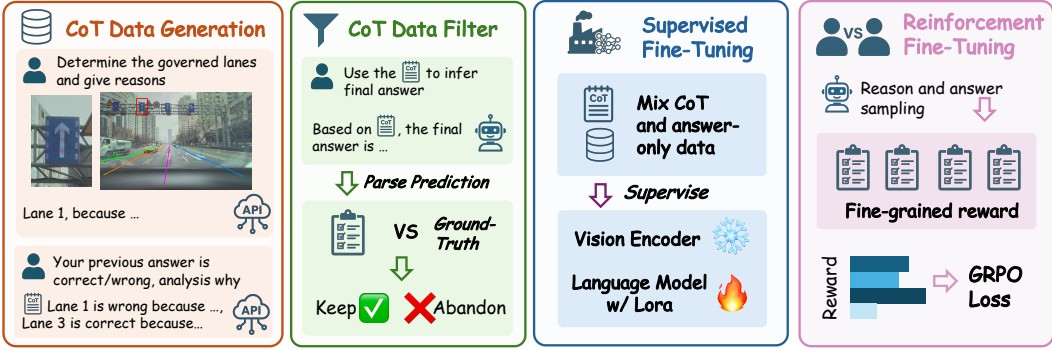

Figure 2: **Overview of our framework.** The workflow consists of four stages: (i) generating candidate CoT examples with an LLM API; (ii) filtering them to retain only valid (CoT, answer) pairs; (iii) supervised fine-tuning the VLM on the filtered dataset; and (iv) reinforcement fine-tuning with a fine-grained reward that accounts for both rule understanding and rule-lane association.

### 3.1 CoT DATA CURATION PIPELINE

Although prior work (e.g. Kojima et al. (2022); Wei et al. (2022)) has demonstrated that CoT prompting can significantly boost LLM performance on complex reasoning tasks, we find that applying CoT naively to VLMs for traffic rule understanding is substantially more difficult. When prompted to independently infer rule–lane relations, the model frequently generates faulty or incoherent reasoning that lead to incorrect predictions. Even when provided with ground-truth information about the relevant traffic sign and regulated lanes, the model does not reliably produce correct or self-consistent explanations. As shown in Figure 3, the model may generate incorrect or contradict rationale. These failure modes motivate our carefully designed CoT data curation pipeline.

**CoT data generation with a *two-round* strategy.** Based on previous observations, we propose a *two-round* prompting strategy to elicit reliable CoT outputs. In the first round, the model is asked to make an independent prediction without seeing the ground-truth. In the second round, we reveal the correct answer and instruct the model to reflect on its initial prediction, examining its rationale and providing a revised explanation. As illustrated in Figure 4, we use the QWen-VL-Max API to obtain

The traffic sign labelled with red box and cropped out in Image 2 governs lane 8.
**Provide detailed reasoning**: Describe your reasoning step by step, explain how the rule matches (or doesn't match) the traffic sign.
Your reasoning must be logically sound and internally consistent, leading naturally to the final lane prediction. **Do not mention, use or compare against the groundtruth or any previous prediction.**

In Image 3 (Bird's-Eye View), the red star marking the "Target Traffic Sign" is positioned directly above the purple-colored lane centerline labeled "8". This spatial alignment indicates that the sign governs the lane immediately beneath it — lane 8.
In Image 1 (Front View), the sign is mounted on a pole to the right side of the road, but its placement and orientation suggest it applies to the leftmost lane from the driver's perspective — which is lane 8 (purple). No other lanes (e.g., 5, 6, 7) are visually or geometrically aligned with the sign. The BEV confirms no shared jurisdiction or merging geometry extending the sign's effect beyond lane 8.

Figure 3: Example of incorrect and self-contradictory CoT reasoning generated by a VLM. When prompted with the rule description and governed lane indices, the model first asserts that the red-boxed traffic sign is "directly above" lane 8, even though they are far apart and located on opposite sides of the image. It then infers that "innermost" corresponds to the leftmost lane, which is correct but contradicts the earlier adjacency-based rationale. Such inconsistencies illustrate that the model's explanations are not causally faithful to the visual evidence, even when the final lane prediction happens to be correct.

(CoT, answer) pairs for traffic-rule understanding. For each scene, we provide three complementary views as inputs: *1)* a front-view image where the target sign and candidate lanes are highlighted; *2)* a cropped sign image; and *3)* a BEV rendering overlaying lane polylines and the sign location. These images collectively supply sufficient 3D lane geometry, front-view semantics, and a clear sign depiction. We also include annotated rule attributes (*e.g.*, category, scope, effective time, applicable lane types) as textual context.

We then employ a *two-round* prompting scheme. In *Round 1*, the model is asked to identify the governed lanes and provide a step-by-step rationale. In *Round 2*, the ground-truth governed lanes are revealed, and the model is instructed to compare them with its initial prediction, analyze the causes of agreement or discrepancy, and revise its explanation. Finally, the model produces an *independent, self-consistent* chain of thought that justifies why the rule applies (or not) to each candidate lane, and reissues the final decision based on this rationale. As shown in Figure 4, the model often makes incorrect predictions when reasoning directly. Once the correct governed lanes are revealed in Round 2, however, it can analyze the cause of its earlier error, reconstruct a reasoning process that leads to the correct outcome, and avoid the internal contradictions illustrated in Figure 3. This demonstrates the effectiveness of the two-round scheme in eliciting causally faithful rationales.

**VLM-based Filtering** Even with the *two-round* strategy, not all generated (CoT, answer) pairs are reliable or consistent. To further filter out low-quality cases, we adopt a VLM-based validation step. Specifically, we feed the generated CoT into a separate VLM as a "judge" model, prompting it to re-derive the final answer from the given CoT. If the judge's answer matches the original ground truth, we keep the sample; otherwise, we discard it, as shown in the second part of Figure 2. This validation ensures that only CoT examples whose reasoning logically leads to the correct answer are retained, thereby reducing noise and promoting higher-quality CoT supervision for training.

## 3.2 TWO-STAGE TRAINING

After constructing the CoT-enhanced dataset, we adopt a two-stage training pipeline to progressively improve the model's reasoning and rule–lane association abilities Qian et al. (2025); Yoshihara et al. (2025); Chen et al. (2025a); Zhang et al. (2025). In Section 3.2.1, we merge the original dataset with generated CoT examples for supervised fine-tuning, initializing the VLM with instruction-guided reasoning ability. In Section 3.2.2, we introduce an answer-grounded reward over multiple reasoning samples for each query together with a GRPO Shao et al. (2024) loss to further refine decision accuracy.

Figure 4: **Examples of the CoT generation pipeline.** We illustrate two examples from our CoT data generation process. In each case, the red arrows ← mark the lanes incorrectly inferred by the LLM in the first prediction, while the green arrows ← indicate the corrected lanes after the second round of reasoning. (Arrows are for illustration only and are not included in the images shown to the VLM.) For the first example, the cropped sign image contains Chinese text meaning "WorkDays. 17:00-19:00. Main road. The innermost lane is designated as a bus-only lane." For the second example, the cropped sign image contains Chinese text meaning "Carriageway".

### 3.2.1 SFT WARM-UP FOR REASONING

In the first stage, we perform Supervised Fine-Tuning (SFT) to enable the model to respond to prompts, carry out reasoning, and produce final answers. We adopt a *mixed prompt strategy*, alternating between:

- CoT-style prompts (*e.g.*, "Think step-by-step: analyze the rule, then relate it to each lane, and finally give the answer.") when training on CoT-augmented data, and
- Direct prompts (*i.e.*, request only the final answer without intermediate reasoning) when training on answer-only data.

This hybrid supervision enables the model to learn from both answer-only examples and CoT-augmented dataset (generated from our pipeline in Section 3.1). The CoT prompts guide the model to internalize reasoning flows, while the direct prompts preserve its ability to deliver concise outputs when needed. SFT serves as an effective warm start: it endows the model with foundational reasoning skills and aligns it with instruction-following. However, SFT often falls short of achieving high accuracy on its own, hence the necessity of a subsequent reinforcement learning stage.

### 3.2.2 RFT ENHANCEMENT VIA GRPO

To further improve the model beyond imitation learning, we adopt Reinforcement Learning Fine-Tuning (RFT) with *Group Relative Policy Optimization (GRPO)*. GRPO Shao et al. (2024) is designed to optimize rewards without relying on a separate critic model, which makes it both efficient and robust for reasoning-oriented tasks.

Concretely, for each input $x$, the current policy $\pi_\theta$ generates a group of $n$ candidate responses $\{y_i\}_{i=1}^n$. Each candidate is assigned a scalar reward $r(y_i)$. Within the group, we compute the empirical mean $\bar{r} = \frac{1}{n} \sum_i r(y_i)$ and standard deviation $\sigma_r$. The GRPO objective encourages the

model to favor responses that achieve above-average rewards relative to the group baseline:

$$\mathcal{L}_{\text{GRPO}}(\theta) = -\mathbb{E}_{y_i \sim \pi_\theta(\cdot|x)} \left[ \log \pi_\theta(y_i \mid x) \frac{r(y_i) - \bar{r}}{\sigma_r} \right]. \quad (1)$$

Intuitively, GRPO compares responses within a sampled group, up-weighting those that score above the group reference and down-weighting those below; this group-relative weighting steers the policy toward higher-reward outputs. For MapDR Chang et al. (2025), we adopt a *fine-grained* reward that scores (i) semantic understanding of the rule and (ii) rule–lane association (Alg. 1) jointly. Compared to binary correctness signals, our rewards provide *denser credit* under partial correctness and yield more informative gradients This design is aligned with recent evidence that finer-resolution feedback improves optimization and downstream quality in RLHF-style training Wu et al. (2023b); Lightman et al. (2023). Within our GRPO-based RFT stage, such granularity further enables effective group-relative preference updates that bias the policy toward higher-reward responses Shao et al. (2024).

---

**Algorithm 1** MAPDR Reward

1: **Parse & validate:** read and parse rule and candidate lane set from prediction; **if fail return** 0.
2: **Rule understanding:** $S_{\text{understand}} = \frac{N_{\text{match}}}{N_{\text{attr}}}$, where $N_{\text{match}}$ is the number of matched rule attributes and $N_{\text{attr}}$ is the total attribute count.
3: **Rule–lane association:** $S_{\text{relation}} = \frac{N_{\text{correct}}}{N_{\text{pred}} + N_{\text{gt}} - N_{\text{correct}}}$, where $N_{\text{pred}}$ is the number of predicted associations, $N_{\text{gt}}$ the ground-truth associations, and $N_{\text{correct}}$ the correctly matched ones (IoU over lane associations).
4: **Final reward:** $R = \frac{1}{2}\left(S_{\text{understand}} + S_{\text{relation}}\right)$; **return** $R$.

---

## 4 EXPERIMENTAL RESULTS

### 4.1 DATASET AND METRICS

**Dataset.** We evaluate on **MapDR** Chang et al. (2025), the first public large scale benchmark collected from real-world traffic scenes that couples traffic signs with locally perceived vectorized HD maps. MapDR provides vectorized lane and rule attribute annotations (*e.g.*, lane type, allowed transport class, effective date/time, speed-limit zone), together with lane correspondences, enabling rule–lane association. In total, MapDR contains 11060 training samples and 1076 validation samples.

**Evaluation Metrics.** Following MapDR Chang et al. (2025), we report Rule Extraction (R.E.) metrics — precision $P_{\text{R.E.}}$ and recall $R_{\text{R.E.}}$ — to measure how well the model extracts the traffic rules themselves. We also compute an end-to-end F1 score, which evaluates the consistency between the predicted graph $\hat{G} = (R \cup L, \hat{E})$ and the ground truth $G = (R \cup L, E)$. For full details and formal definitions, refer to the original paper Chang et al. (2025).

$$P_{\text{RE}} = \frac{|\hat{R} \cap R|}{|\hat{R}|}, \quad R_{\text{RE}} = \frac{|\hat{R} \cap R|}{|R|}, \quad P_{\text{ALL}} = \frac{|\hat{G}^s \cap G^s|}{|\hat{G}^s|}, \quad R_{\text{ALL}} = \frac{|\hat{G}^s \cap G^s|}{|G^s|}, \quad F_1 = \frac{2\,P_{\text{ALL}}\,R_{\text{ALL}}}{P_{\text{ALL}} + R_{\text{ALL}}}. \quad (2)$$

### 4.2 IMPLEMENTATION DETAILS

#### 4.2.1 TRAINING SETUP

**CoT data generation and filtering.** We first synthesize chain-of-thought (CoT) rationales via the `qwen-vl-max` API, and then filter them using **Qwen2-VL-72B-Instruct**, discarding samples that misinterpret sign content or mismatch lane attribution. From 11060 initial "answer-only" samples, we retain 4517 valid (CoT, answer) pairs for training.

**Backbone and LoRA.** Following RuleVLM Chang et al. (2025), we adopt **Qwen-VL-Chat (9.6B)** as the pretrained backbone and use the **MEE** module to encode lane vectors and fuse rule embeddings with vector features for rule–lane reasoning, ensuring a controlled comparison in model size and architecture. For parameter-efficient fine-tuning, we employ **LoRA** Hu et al. (2022) with rank $r = 64$, scaling $\alpha = 16$, and dropout 0.05. Adapters are attached to `attn.c_proj`, `w2`, `w1`, and `c_attn`, following common Qwen-VL practice.[2]

---

[2]Official Qwen-VL finetuning script: `https://github.com/QwenLM/Qwen-VL/blob/master/finetune.py#L59`.

**Supervised Fine-Tuning (SFT).** We first warm up the model with supervised fine-tuning. Training is conducted using AdamW Loshchilov & Hutter (2019) (weight decay $0.1$) and a cosine learning-rate schedule Loshchilov & Hutter (2017) peaking at $1 \times 10^{-5}$, with gradient clipping at $0.1$. We set the batch size to 64 and train for 10 epochs in this stage. To accommodate memory demands and scale efficiently, we employ DeepSpeed ZeRO-2 Rajbhandari et al. (2020).

**Reinforcement Fine-Tuning (GRPO).** After SFT, we further refine the model with GRPO Shao et al. (2024). We train for one epoch with batch size 32, using weight decay $0.02$, warmup ratio $0.01$, and gradient clipping at $0.1$. For each prompt, we sample $8$ candidate generations per group. The LoRA adapters and ZeRO-2 setup from SFT remain active during reinforcement training.

## 4.3 MAIN RESULTS

Table 1: **Overall evaluation on MapDR.** We follow RuleVLM's protocol and metrics Chang et al. (2025): *Rule Extraction (R.E.)* measures the precision/recall of recovered driving rules, and *Rule–Lane Correspondence Reasoning (C.R.)* measures the precision/recall of rule–lane matches. Under this setup, our method achieves higher overall score.

| Model | Type | R.E. | | C.R. | | Overall | | |
|---|---|---|---|---|---|---|---|---|
| | | $P_{R.E.}(\%)$ | $R_{R.E.}(\%)$ | $P_{C.R.}(\%)$ | $R_{C.R.}(\%)$ | $P_{all}(\%)$ | $R_{all}(\%)$ | $F1\ Score$ |
| Heuristic | | 18.01 | 11.51 | 33.05 | 17.99 | 5.01 | 2.73 | 0.035 |
| ALBEF-BERT | Modular | 75.78 | 57.56 | 4.14 | 17.25 | 0.24 | 0.78 | 0.003 |
| VLE-MEE | | 76.67 | 74.54 | 78.05 | 82.16 | 63.35 | 67.37 | 0.653 |
| Qwen-VL(TextPrompt) | | 42.21 | 41.09 | – | – | 8.39 | 8.17 | 0.083 |
| Qwen-VL(VisualPrompt) | End-to-End | 89.29 | 89.50 | – | – | 39.14 | 39.23 | 0.392 |
| RuleVLM Chang et al. (2025) | | 89.28 | 89.44 | – | – | 64.16 | 64.28 | 0.642 |
| Ours | | 87.71 | 86.91 | – | – | 72.00 | 72.56 | 0.723 |

As shown in Table 1, our method achieves the best overall F1 score (0.723), outperforming both modular baselines and end-to-end models. Interestingly, we observe that the performance on rule extraction (R.E.) is slightly lower than RuleVLM (87.71 vs. 89.28), while the overall score is higher (0.723 vs. 0.642). This discrepancy can be explained by the evaluation protocol in RuleVLM Chang et al. (2025), where the overall metric jointly considers both rule extraction (R.E.) and correspondence reasoning (C.R.). A model that extracts rules with high precision but fails to align them correctly with lanes may achieve strong R.E. scores yet struggle in the overall evaluation. Conversely, our method, despite minor losses in attribute-level extraction accuracy, achieves substantially stronger correspondence reasoning, leading to higher consistency between extracted rules and governed lanes. These results suggest that through the introduction of CoT, our method achieves a notable improvement specifically in rule-lane relation reasoning; this underscores how enhancing reasoning is crucial for accurate traffic rule understanding.

## 4.4 ABLATION STUDIES

Table 2: **Ablation on data and training strategy.** We vary the *training data* (answer only, CoT only, mixed) and the *training strategy* (SFT vs. SFT + GRPO). Metrics follow the MapDR protocol ($P_{\text{all}}$, $R_{\text{all}}$, F1) Chang et al. (2025). GRPO consistently boosts performance over SFT alone, and the mixed answer + CoT setup attains the best overall F1.

| Configuration | SFT | | | SFT + GRPO | | |
|---|---|---|---|---|---|---|
| | $P_{\text{all}}(\%)$ | $R_{\text{all}}(\%)$ | F1 score | $P_{\text{all}}(\%)$ | $R_{\text{all}}(\%)$ | F1 score |
| Answer only | 64.16 | 64.28 | 0.642 | 67.36 | 66.75 | 0.671 |
| CoT only | 52.64 | 50.93 | 0.518 | 68.28 | 70.56 | 0.694 |
| Mix | 57.60 | 55.33 | 0.564 | 72.00 | 72.56 | 0.723 |

Table 2 varies the *training data* (answer only, CoT only, mixed) and the *training strategy* (SFT vs. SFT + GRPO). (1) **Answer only data, SFT** refers the RuleVLM setting Chang et al. (2025) and we simply use their results; (2) **Answer only data, SFT+GRPO** further fine-tunes the RuleVLM-pretrained model with GRPO and yields clear gains in $P_{\text{all}}$, $R_{\text{all}}$, and F1, indicating that under our *fine-grained* reward and group-relative comparison—GRPO improves the end objective beyond simple imitation. (3) **CoT only data, SFT** underperforms, indicating that CoT-only at the SFT stage does not sufficiently cover the final-answer distribution Liu et al. (2023). However, **CoT only data,**

Table 3: **Comparison between coarse and fine grained reward.**

| Reward type | $P_{\text{all}}(\%)$ | $R_{\text{all}}(\%)$ | **F1** |
|---|---|---|---|
| Coarse | 70.84 | 71.44 | 0.711 |
| Fine | 72.00 | 72.56 | 0.723 |

**SFT+GRPO** recovers substantially, suggesting that a model endowed with CoT capability benefits more from GRPO than a non-CoT counterpart; this underscores the task's demand for stronger reasoning and validates our use of CoT. (4) Finally, **Mix (answer+CoT) data, SFT+GRPO** attains the best overall F1, showing that combining direct-answer supervision with CoT rationales provides complementary signals, and GRPO effectively leverages this mix to bias the policy toward higher-reward responses.

**Ablation on fine grained rewards.** For comparison, we define a coarse reward: +0.5 if the rule semantics are correct, +0.5 if the rule–lane relation is correct (thus $1.0$ if both are correct), and $0$ if the output is unparsable/invalid. Unlike our fine-grained scheme, this piecewise–binary signal provides sparse credit and weaker gradients. As shown in Table 3, the fine-grained reward improves precision, recall, and F1, confirming its effectiveness.

### 4.5 QUALITATIVE COMPARISON

We present four qualitative comparisons with RuleVLM in Figure 5. While RuleVLM directly predicts lane indices, our model first reasons (interpreting the sign, then aligning with spatial geometry), and only then outputs the final lanes. In all four cases, our method successfully parses sign contents in the cropped view and fuses those with BEV/FV spatial cues to correctly associate the sign with the appropriate lane IDs, whereas RuleVLM fails. These results underscore the value of explicit reasoning and our method delivers both stronger interpretability and robustness.

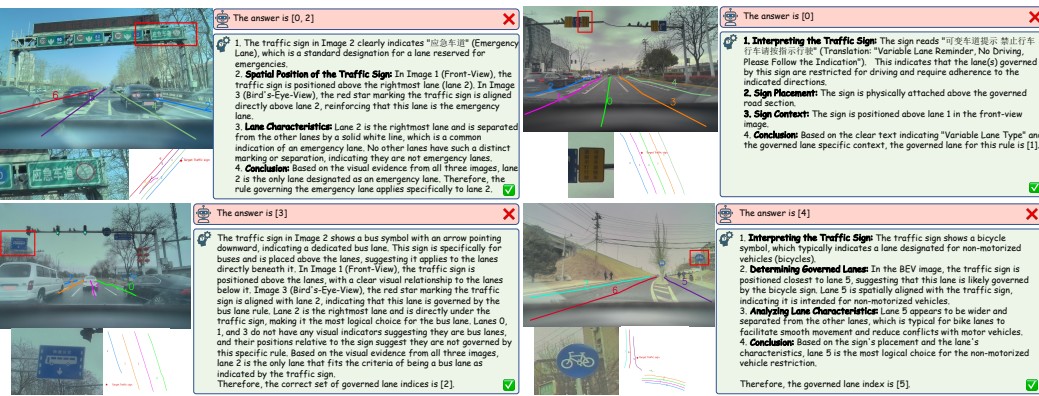

Figure 5: **Qualitative comparison with RuleVLM.** Outputs from RuleVLM are annotated with 🖥, while ours are marked with 🤖. A ❌ denotes a wrong prediction and ✅ a correct one. In these cases, our method follows a reasoning process—interpreting the sign, aligning spatial cues, and then selecting the lane—resulting in correct predictions, whereas RuleVLM fails, highlighting both higher accuracy and stronger interpretability through reasoning.

## 5 CONCLUSION

We tackle *traffic rule understanding* on MapDR by introducing chain-of-thought (CoT). Concretely, we build a CoT data curation pipeline, then adopt a two-stage training scheme: an SFT warm-up on mixed (answer+CoT) data, followed by GRPO-based RFT driven by our fine-grained reward that jointly scores rule semantics and rule–lane association. This combination yields consistent improvements over RuleVLM, highlighting the effectiveness of reasoning in this task.

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

## A   APPENDIX

### A.1   LLM USAGE

We use the LLM only for polishing text and generating the small illustrative icons used in figures.

