# OpenReview forum: "Reasoning to Regulate: Chain-of-Thought for Traffic Rule Understanding"
_ICLR.cc/2026/Conference — Submitted to ICLR 2026_

### Official Review · Reviewer_hRyd · 2025-10-27

**Soundness:** 2
**Presentation:** 3
**Contribution:** 1
**Rating:** 2
**Confidence:** 4

**Summary:**

This paper proposes to introduce CoT into traffic sign understanding. A pipeline of CoT data collection, verification and application is designed, and deployment of such a method results in a better performance in traffic sign understanding.

**Strengths:**

1. The data collection (automatic pipeline) is carefully designed. While there could be problems (weakness 3), these can at least help to reduce human labor and improve the acceptance rate for auto-generated data.
2. The authors make use of LLM's comprehensive abilities in data collection and data judgement, to provide data (though quality hard to quantitatively evaluate) helpful to downstream tasks like traffic sign understanding, which could be beneficial in the engineering level.

**Weaknesses:**

1. The novelty might be limited. It seems that CoT is nearly a standard operation for any reasoning LLM, including those used as base models for driving LLMs. Furthermore, it seems that few labeled reasoning data are introduced, which to me further questions the contribution of this work in improving CoT reasoning.
2. The motivation is questionable. For most traffic signs, they are standard so understanding them via VLM might not be beneficial to the overall accuracy, considering VLM's performance compared to direct perceptions of signs in specific patterns. For rare non-standard traffic signs, they are normally presented with short and clear text, where OCR-then-LLM might be much more straightforward. Therefore, I recommend the authors to reconsider the necessity of motivation.
3. The quality of the CoT data should be human verified. These data serve as the textbook for the VLM to generalize, so their accuracy is very important. Without human verification or at least human evaluations of their accuracy, it is hard to evaluate their contributions in downstream tasks.

**Questions:**

Please see the weakness part.

**Details Of Ethics Concerns:**

No ethics concerns found.

---

> ### Comment · Reviewer_U4eg · 2025-11-26
>
> Hi Reviewer hRyd,
>
> Actually understanding traffic signs with VLM have peripheral effects on scene understanding (or action understanding). But the reviewer agrees with Reviewer hRyd that with current motivation, OCR-then-LLM can be a good baseline.

---

> > ### Comment · Reviewer_hRyd · 2025-11-26
> >
> > Hi reviewer U4eg,
> >
> > Thanks for your opinion! Intuitively I agree that using reasoning LLM to understand traffic signs can be a secondary backup after pattern recognition and OCR-then-LLM. However, in this case, I think the advantage of using the reasoning LLM should be clearly provided by comparisons to justify the motivation. I agree that more baseline is needed to strengthen the motivation.
> >
> > Thanks,
> > Reviewer hRyd

---

### Official Review · Reviewer_fnU6 · 2025-10-29

**Soundness:** 2
**Presentation:** 2
**Contribution:** 2
**Rating:** 4
**Confidence:** 4

**Summary:**

This paper introduces CoT reasoning to the task of traffic rule understanding and lane association using VLMs. It proposes a pipeline to generate and filter CoT data, followed by a two-stage training process (SFT + GRPO) with fine-grained rewards, achieving improved accuracy and interpretability on the MapDR benchmark.

**Strengths:**

The paper's key contribution is integrating explicit CoT reasoning into the challenging task of mapping traffic rules to specific lanes, moving beyond simpler end-to-end prediction. Its strengths include a novel, scalable CoT data curation pipeline featuring a two-round generation strategy and VLM-based filtering . The two-stage training (SFT + GRPO) with fine-grained rewards is well-structured and demonstrates effectiveness. Results show a significant boost in the overall F1 score, primarily driven by improved rule-lane correspondence reasoning, supporting the value of the reasoning-focused approach.

**Weaknesses:**

While the paper aims to unify rule understanding and lane association via CoT, the approach still addresses two distinct components (rule extraction and correspondence reasoning), as reflected in the metrics and reward structure . The CoT acts as a bridge, but its effectiveness in creating a truly synergistic solution versus a sequential execution of sub-tasks could be further explored. Additionally, the reliance on external VLM/LLM APIs for CoT generation and filtering raises concerns about dependency and potential bias propagation. The slight decrease in Rule Extraction performance versus the baseline also needs attention.

**Questions:**

1. The CoT generation and filtering depend on specific Qwen APIs. How sensitive is the data quality and subsequent model performance to the choice of these external models without finetuning themselves?
2. Table 1 shows improved Correspondence Reasoning but slightly lower Rule Extraction scores compared to RuleVLM. Does the CoT approach inherently cause a trade-off, potentially focusing reasoning resources on spatial association at the expense of fine-grained rule attribute extraction?
3. Need more elaboration on the VLM-based filter, how reliable it is? What kinds of incorrect reasoning does it successfully catch, and are there common failure modes where faulty CoT might still pass validation?

---

### Official Review · Reviewer_o6yi · 2025-11-01

**Soundness:** 2
**Presentation:** 2
**Contribution:** 2
**Rating:** 2
**Confidence:** 4

**Summary:**

The authors propose a new reasoning data augmentation pipeline that claims to improve the rule-to-lane understanding of current models.

**Strengths:**

The paper curates a reasoning data augmentation pipeline that seemingly demonstrates its potential usefulness towards traffic scene understanding. However, the paper lacks of significant evidence and experimental details to support its arguments.

**Weaknesses:**

While the paper presents a new reasoning-based framework for traffic rule understanding, several aspects of its experimental reporting and analysis remain insufficiently detailed. First, the paper does not provide enough statistics or transparency regarding the training configuration of the vision-language models, such as dataset split ratios, number of training samples after filtering, hyperparameter choices, or computational resources used. This lack of detail makes it difficult to assess the reproducibility and scalability of the proposed approach. Second, while the authors emphasize the construction of a high-quality Chain-of-Thought (CoT) dataset through multi-stage filtering and verification, the paper does not clearly quantify how data quality improvements translate to model performance. For instance, it remains unclear how much the filtering process or rationale correctness directly contributes to the observed performance gains. Finally, the paper provides limited qualitative analysis to illustrate the strengths of the proposed method. Although some examples of reasoning outputs are shown, these are sparse and lack comparative visualization against baseline models, making it difficult to understand how the model’s reasoning improves interpretability or decision accuracy in specific traffic scenarios. Together, these gaps leave the reader uncertain about the robustness and practical advantages of the proposed reasoning framework.

**Questions:**

Refer to weaknesses.

---

### Official Review · Reviewer_U4eg · 2025-11-01

**Soundness:** 3
**Presentation:** 3
**Contribution:** 3
**Rating:** 6
**Confidence:** 3

**Summary:**

This paper proposes a VLM with chain-of-thought plus two stage training to map traffic rules to governed lanes on MapDR. The problem is that existing models like RuleVLM directly generate rule lane outputs but cannot reason over sign attributes and lane topology together. Experiments like Table 1 show that the proposed method gets 72.3 F1 while RuleVLM gets 64.2 proving explicit reasoning improves lane level correspondence on MapDR.

**Strengths:**

- The reviewer finds the proposed idea to insert curated CoT and GRPO reward into MapDR style rule lane association to be interesting.
- Interestingly, rule extraction precision drops from 89.28 to 87.71 but overall F1 still rises which means better reasoning beats pure extraction.
- The experiments in Table 1, Table 2 and Figure 1 clearly show gains from CoT curation and from SFT plus GRPO on the same backbone.
- Writing is easy to follow.

**Weaknesses:**

- The CoT filtering stage in Section 3.1 keeps 4517 of 11060 samples but does not report recall or noise rate so reliability is unclear. Authors can clarify this.
- The experiments do not add a simple geometry only lane nearest to sign baseline or a sign text only baseline although lines 324 to 335 explain GRPO reward on those parts so these should be easy to report.
- A minor typo - Line 209: contradict -> contradictory

**Questions:**

- In Table 1, "Ours" has lower PR.E than RuleVLM but higher overall F1. Can you decompose gains into rule understanding and rule lane association?
- Section 3.1 says a two round LLM generates and revises CoT and a VLM verifier filters but the exact prompt templates and pass thresholds are not given. Can you please add this to the paper?

---

### Meta-Review · Area_Chair_yKzi · 2026-01-07

**Summary:**

This paper proposes to introduce CoT into traffic sign understanding and lane association. A pipeline of CoT data collection, verification and application is designed, and deployment of such a method results in a better performance in traffic sign understanding. It uses a two-stage training process (SFT + GRPO) and is evaluated on the MapDR benchmark.

Reviewers expressed concerns including:
(1) Limited novelty (e.g. CoT standard to 'reasoning' LLM base models) and motivation, given standardization of traffic signs. A reviewer recommends OCR-then-LLM, however, this is not necessarily reliable over full end-to-end VLM approaches.
(2) Quality of CoT data without human verification and impact on model performance.
(3) Unclear synergy between rule extraction and correspondence reasoning tasks.
(4) Biases of external VLM/LLM APIs without finetuning.
(5) Decrease in Rule Extraction performance relative to baseline.
(6) Training configuration details of VLMs (e.g. dataset splits, training samples, hyperparameters, computational resources, thresholds) which may affect reproducibility.

**Reviewer Concerns:**

There was no rebuttal posted by the author.

**Reviewer Scores:**

hRyd: keep score (2).
fnU6: keep score (4).
o6yi: keep score (2).
U4eg: keep score (6).

---

### Decision · Program_Chairs · 2026-01-26

Reject